# Comparison of Outcomes between Unilateral Biportal Endoscopic and Percutaneous Posterior Endoscopic Cervical Keyhole Surgeries

**DOI:** 10.3390/medicina59030437

**Published:** 2023-02-23

**Authors:** Dong Wang, Jinchao Xu, Chengyue Zhu, Wei Zhang, Hao Pan

**Affiliations:** 1Department of Orthopaedics, Hangzhou Traditional Chinese Medicine Hospital Affiliated to Zhejiang Chinese Medical University, Tiyuchang Road No. 453, Hangzhou 310007, China; 2Department of Orthopaedics, Hangzhou Dingqiao Hospital, Huanding Road No. 1630, Hangzhou 310021, China; 3Sports Medicine Department, The Second People’s Hospital of Fujian University of Traditional Chinese Medicine, Wusi Road No. 282, Fuzhou 350001, China

**Keywords:** unilateral biportal endoscopy, full endoscopy, posterior cervical foraminotomy, cervical spondylotic radiculopathy

## Abstract

*Objective*: The purpose of this study was to compare the clinical and radiological outcomes of unilateral biportal endoscopic (UBE) and percutaneous posterior endoscopic cervical discectomy (PE) keyhole surgeries. *Methods*: Patients diagnosed with cervical spondylotic radiculopathy (CSR) treated by UBE or PE keyhole surgery from May 2017 to April 2020 were retrospectively analyzed. The length of incision, fluoroscopic time, postoperative hospital stay, and total cost were compared. The clinical efficacy was assessed using a visual analog scale (VAS), neck disability index (NDI), and modified MacNab criteria. Moreover, the C2-7 Cobb’s angle, range of motion (ROM), intervertebral height, vertebral horizontal displacement, and angular displacement of the surgical segment were measured. *Results*: A total of 154 patients were enrolled, including 89 patients in the UBE group and 65 patients in the PE group, with a follow-up period of 24–32 months. Compared with PE surgery, UBE surgery required shorter fluoroscopic times (6.76 ± 1.09 vs. 8.31 ± 1.10 s) and operation times (77.48 ± 17.37 vs. 84.92 ± 21.97 min) but led to higher total hospitalization costs and longer incisions. No significant differences were observed in the postoperative hospital stay, bleeding volume, VAS score, NDI score, effective rate, or complication rate between the UBE and PE groups. Both the C2-7 Cobb’s angle and ROM increased significantly after surgery, with no significant differences between groups. There were no significant differences between intervertebral height, vertebral horizontal displacement, and angular displacement of the surgical segment at different times. *Conclusions*: Both UBE and PE surgeries in the treatment of CSR were effective and similar after 24 months. The fluoroscopic and operation times of UBE were shorter than those of PE.

## 1. Introduction

Cervical spondylotic radiculopathy (CSR), a common disease treated with spinal surgery, refers to a series of symptoms and signs resulting from mechanical compression and inflammatory stimuli associated with nerve roots due to degenerative spinal changes, such as intervertebral disc herniation and articular process hyperplasia [1]. Many patients need to undergo surgery to relieve their symptoms after failed conservative treatment [2].

With the development of minimally invasive surgery and endoscopic visualization, percutaneous posterior endoscopy (PE), as one of the minimally invasive decompression techniques, has achieved significant clinical results in the treatment of CSR and is considered to have the advantages of shorter hospital stay, less tissue damage, less blood loss, and early functional recovery [3,4]. Unilateral biportal endoscopy (UBE) was first reported by De Antoni [5] in the 1990s. It is worth noting that with the continuous expansion of UBE for the treatment of CSR in recent years, UBE was considered to have more flexible instrument operation, better decompression, and fewer intraoperative fluoroscopy procedures in theory than PE, while continuous saline irrigation made the field of vision clear and the operation safe [6]. Both PE and UBE are used for minimally invasive surgery for CSR, but no studies have been conducted to investigate postoperative pain, functional recovery, and complications in the short term.

The purpose of this comparative study was to compare the clinical and radiological outcomes of UBE and PE. The data of patients with CSR treated in our department from May 2017 to April 2020 were analyzed retrospectively.

## 2. Material and Methods

### 2.1. Patients

Patients who underwent UBE or PE surgery because of CSR from May 2017 to April 2020 in the Department of Orthopedics, Hangzhou Traditional Chinese Medicine Hospital Affiliated to Zhejiang Chinese Medical University were analyzed retrospectively. Other than the surgery and perioperative medication, there was no other redundant treatment and the patients involved in the study did not take additional risks. The institutional review committee of Hangzhou Traditional Chinese Medicine Hospital Affiliated to Zhejiang Chinese Medical University approved this study.

### 2.2. Inclusion and Exclusion Criteria

The following inclusion criteria for patients were used: (1) magnetic resonance imaging (MRI) revealed unilateral foramen stenosis with cervical disc herniation, which was consistent with symptoms such as unilateral upper limb numbness and pain. (2) Patients who underwent UBE or PE because of CSR. (3) Systematic conservative treatment for more than three months was unsuccessful. (4) The age of patients was between 40 and 80 years.

The exclusion criteria included the following: (1) patients presented with severe cervical segmental instability, spondylolisthesis, or severe kyphosis. (2) Patients with incomplete radiological data. (3) The patient presented with cervical spondylotic myelopathy, spinal tumor, and ossification of the posterior longitudinal ligament. (4) The follow-up time was less than 2 years.

### 2.3. Surgical Procedures

#### 2.3.1. UBE

The patient was placed in a prone position with the chest raised under general anesthesia. The operating table was adjusted until the patient’s cervical curvature was straight and parallel to the ground (Figure 1). Taking a right-side approach as an example, the viewing and working portals were located 1 cm proximal and distal to the intersection point of the horizontal line of the intervertebral space and the midline of the lateral mass on an anteroposterior (AP) view, respectively (Figure 2A). The endoscope and radiofrequency probe (BONSS^®^, Taizhou, China) converged at the base of the spinous process and vertebral plate (Figure 2B). The soft tissue was cleared until the V point between the vertebrae was exposed and the extent of decompression was estimated (Figure 3A). Next, the partial ligamentum flavum (LF) was resected to expose the nerve root and dural sac. The axillary and shoulder regions of the nerve root were explored completely, and the herniated disc was removed (Figure 3B). The color and pulsation of the nerve roots returned to normal (Figure 3C). In cases with severe stenosis, a Kirschner wire could be placed into the intervertebral space through an auxiliary portal (the quarterback K portal) to pull and protect the nerve root (Figure 3D–F). After adequate hemostasis, the incision was sutured, and a drainage tube was placed.

The tip and medial part of the superior articular process (SAP) should be removed for complete decompression of the nerve root canal in UBE keyhole surgery (Figure 4A,B). To avoid the adverse influence of excessive removal of the lateral mass joint on cervical stability, the specific decompression range should be confirmed by positioning the lateral and medial edges of the pedicle with a 3 mm burr (Figure 4C,D). In the present study, based on the vertebral horizontal displacement and angular displacement of the surgical segment, it was confirmed that the single-segment cervical stability did not change significantly after UBE and PE surgeries.

#### 2.3.2. PE

The patient was placed in a prone position with the neck slightly bent. The target point was located at the intersection point of the horizontal line of the intervertebral space and the medial line of the lateral mass on AP view (Figure 5A). Under the supervision of the C-arm, the needle was punctured to the bony surface of the target point, and infiltration anesthesia was gradually performed, followed by the placement of the guide wire. The skin, subcutaneous tissue, and deep fascia layer were incised, multistep dilators were inserted, and the working cannula was placed (Figure 5B,C). The soft tissue on the bony surface was cleared to expose the V point under endoscopic vision. Taking the V point as the center, the surrounding bone was removed to enlarge the nerve root canal. Discectomy and facetectomy were performed to completely relax the nerve root. After careful hemostasis, the incision was sutured without a drainage tube.

### 2.4. Outcome Evaluation

#### 2.4.1. Surgical Outcomes

The patients were followed for a minimum of 24 months after surgery. Before surgery and at one week, one month, three months, and six months after surgery, the degree of pain present in the shoulder and upper limbs was assessed using the visual analog scale (VAS). Neurological function was assessed using the neck disability index (NDI). At the last follow-up, surgical efficacy was evaluated using modified MacNab criteria.

#### 2.4.2. Radiological Measurement

Before surgery, six months after surgery, and at the last follow-up, radiological examinations, including cervical anteroposterior, lateral, hyperflexion, and hyperextension radiographs, were assessed and analyzed using Surgimap 2.3.2.1 software (Nemaris^®^, New York, NY, USA). The measurement indicators included the following: (1) C2-7 Cobb’s angle: the angle between the inferior endplate lines of C2 and C7 on the cervical lateral radiograph (Figure 6A). (2) C2-7 ROM: the difference in the angle between the inferior endplate line of C2 and C7 on the cervical hyperflexion and hyperextension radiographs (Figure 6B,C). (3) Intervertebral height: the length of the connecting line of the midpoint of the inferior endplate of the upper vertebrae and the superior endplate of the lower vertebrae of the surgical segment on the cervical lateral radiograph (Figure 6D). (4) Vertebral horizontal displacement: the relative horizontal displacement between the posterior edge of the assessed vertebrae and the adjacent vertebrae (Figure 6E). (5) Vertebral angular displacement: the angle between the inferior endplate line of the two adjacent vertebrae (Figure 6F).

### 2.5. Statistical Analysis

ISM SPSS Statistics v25.0 software (IBM^®^, Foster City, CA, USA) was used for data analysis. Normally distributed quantitative data were expressed as the mean ± SD. Data at different time points before and after surgery were compared by repeated measures analysis of variance. The LSD test was adopted for pairwise comparisons. A *t*-test was used to compare the variables between the UBE and PE groups. The classified data were compared using the Chi-squared test. A *p*-value less than 0.05 was considered to be statistically significant.

## 3. Result

### 3.1. Characteristics of Patients

Based on the inclusion and exclusion criteria, 8 patients were excluded due to loss of follow-up or incomplete data during this study. A total of 154 patients were included, including 89 cases of UBE and 65 cases of PE. The patients were followed for 24–32 months after surgery. The postoperative hospital stays were 6.88 ± 1.92 and 6.38 ± 2.27 days, respectively (*p* = 0.391). The average total hospitalization cost of UBE was 24.09 ± 2.44 thousand RMB, while that of PE was 18.62 ± 2.87 thousand RMB (*p* < 0.001) (Table 1). No significant differences were observed in the basic characteristics, including age, sex, and follow-up duration, between the two groups.

### 3.2. Surgical Outcomes

Surgery-related data are listed in Table 2 and typical cases are shown in Figure 7 and Figure 8. The mean operation time was 77.48 ± 17.37 min in the UBE group and 84.93 ± 21.97 min in the PE group (*p* = 0.020). The mean bleeding volume was 52.00 ± 16.10 mL in the UBE group and 49.48 ± 13.72 mL in the PE group (*p* = 0.165). The average fluoroscopic time was 6.76 ± 1.09 s in the UBE group and 8.31 ± 1.11 s in the PE group, with each fluoroscopy recorded as one second (*p* < 0.001). The average length of the incision required for UBE surgery was 24.52 ± 2.06 mm and that of PE was 11.68 ± 1.88 mm (*p* < 0.001). There was no significant difference between the estimated bleeding volumes of the two groups. Based on the modified MacNab criteria, the effective rates (excellent or good) of the UBE and PE groups were 93.26% and 86.15%, respectively, at the last follow-up (Figure 9), with no significant difference (*p* = 0.142).

In the UBE and PE groups, VAS and NDI scores were significantly improved after surgery and the differences were statistically significant (Table 3). The VAS scores of the neck decreased from 7.93 ± 0.69 and 8.06 ± 0.83 before surgery to 1.72 ± 0.45 and 1.89 ± 0.69 at 12 months following surgery (*p* < 0.05), respectively. The VAS scores of the arms decreased from 6.28 ± 0.94 and 6.35 ± 1.11 before surgery to 1.92 ± 0.80 and 1.86 ± 0.73 at 12 months following surgery (*p* < 0.05), respectively. The NDI scores improved from 35.67 ± 4.24 and 36.25 ± 3.40 before surgery to 14.58 ± 3.09 and 15.18 ± 3.23 at 12 months after surgery, respectively. There were no significant differences in VAS and NDI scores between the two groups (Table 3).

### 3.3. Complications

Complications occurred in 4 patients in the PE group and 3 patients in the UBE group (Table 2). Dural tears were observed in 1 patient in the PE group and 2 patients in the UBE group, but two patients did not develop clinical symptoms after surgery. The patient with dural tears in the UBE group developed postoperative neck stiffness and headache. Symptoms improved significantly on the second day after surgery after maintaining a dorsal elevated position, oxygen administration, and intravenous mannitol infusion. There was 1 case of nerve root injury in the PE group, which manifested as paralysis of the thumb dorsi muscle. The nucleus pulposus was not completely removed in 3 cases and patients had residual upper limb pain after surgery, which was relieved after 1 to 3 months of conservative treatment.

### 3.4. Radiological Measurement

Both the C2-7 Cobb’s angle and ROM increased significantly after surgery. The Cobb’s angle increased from 16.87 ± 2.89° and 16.66 ± 2.70° before surgery to 24.46 ± 3.94° and 23.53 ± 6.02° at the last follow-up, respectively. The C2-7 ROM increased from 22.60 ± 7.59° and 22.08 ± 8.23° to 27.57 ± 9.45° and 26.42 ± 9.62°, respectively. There were no significant differences in the C2-7 Cobb’s angle or ROM between the UBE and PE groups (Table 4). No significant differences were noticed in the preoperative and postoperative intervertebral height, vertebral horizontal displacement, and angular displacement of the surgical segment between the groups.

## 4. Discussion

### 4.1. Technical Development and Characteristics

In recent years, posterior cervical endoscopic surgery has developed into a safe and effective minimally invasive spinal surgical technique. PE reduces the complications related to the anterior approach and fusion surgery, while retaining the range of motion of the surgical segment, thus reducing the incidence of ASD [7,8]. PE had almost no effect on the stress distribution, intervertebral disc pressure, and ROM of operative and adjacent segments [9,10]. As a uniportal long-axis technique, PE can only be performed along the working channel. Thus, the movement of instruments is restricted such that it is difficult to adequately expose the surgical target [11,12].

UBE technology evolved from arthroscopy and has the advantages of wide vision and flexible operation [5,13]. Park et al. [14] reported 13 cases of CSR treated with UBE with significant improvement of VAS and NDI scores after follow-up for 12–18 months. In our study, the operation time of UBE was shorter than that of PE, which may be due to the increased work efficiency resulting from the larger working space and better flexibility of instruments during UBE surgery.

### 4.2. Effectiveness

In this study, the follow-up results revealed that compared with the preoperative measurements, the cervical Cobb’s angle and ROM were significantly improved at the last follow-up in both the UBE and PE groups. Studies have indicated that improvements of cervical Cobb’s angle and ROM were achieved primarily through the removal of the nucleus pulposus, relief of symptoms, and preservation of the posterior cervical structures [15,16]. In this study, the VAS and NDI scores improved significantly after surgery in both the UBE and PE groups, which confirmed the preservation of the posterior cervical structure and reliable curative effect of UBE and PE. However, there were no significant differences in the cervical Cobb’s angle, ROM, or the VAS and NDI scores between the UBE and PE groups. This indicated that although the technical characteristics of UBE and PE were different, there was no significant difference in the improvement of pain and function of the two techniques during the two-year follow-up.

### 4.3. Cervical Stability

An increased cervical ROM increases the stress load on intervertebral disc tissues, thus increasing the risk of recurrence and degeneration [17]. In addition, when the articular process is insufficiently exposed in the outward orientation, extra removal of the articular process is needed, which adversely affects posterior stability [18]. However, no axial pain occurred during the follow-up period of the patients in this study. It was previously reported that postoperative axial neck and shoulder pain was primarily related to the loss of intervertebral height, cervical instability, and cervical paravertebral muscle atrophy [19]. There were three possible reasons for the decreased incidence rate of postoperative axial pain observed in this study. First, the herniated nucleus pulposus was removed without destroying the original intervertebral disc tissues, which allowed the residual disc tissues to retain their original function and preserve the intervertebral height. Second, UBE and PE had little interference with the posterior cervical muscles. Therefore, the paravertebral muscles, ligaments around the articular process, and the dorsal branch of the nerve root were kept intact to the greatest possible extent. Third, UBE and PE preserved the bony structure of the articular process as much as possible and improved cervical stability [6,20].

### 4.4. Precautions of Complications

During spinal endoscopic surgery, continuous saline perfusion might cause increased intracranial pressure, which leads to initial symptoms of neck pain and stiffness [21]. If the condition progresses, it may manifest as neurological dysfunction, such as headache, visual impairment, and even seizure [22,23]. This syndrome, known as myeloid hypertension [24], is often detected during PE surgery because of local anesthesia. Myeloid hypertension is often secondary to an intraoperative dural tear, the most common complication of endoscopic spine surgery [25], and is thought to be caused by direct compression of the conus spinal cord by reverse infusion of fluid into the subarachnoid space through dural rupture [26]. Saline should flow smoothly during surgery by transversely cutting the deep fascia, applying a semicannula, or adding a third portal [27,28]. Dural tears should be avoided with care during the surgery. Once it occurs during the operation, perfusion pressure should be reduced immediately, and the surgery should be completed as soon as possible.

### 4.5. Radiation Exposure

The study by Merter et al. [29] indicated that the radiation exposure of percutaneous endoscopic surgery was greater than that of UBE or microendoscopy in lumbar discectomy surgery. In addition, as disc herniations become more complex and surgical difficulties increase, the time of fluoroscopic applications increases [30]. In our study, the fluoroscopic time of PE, which may be a less invasive method with a shorter length of incision, was also longer. Compared with UBE, lateral fluoroscopy was more frequent in PE surgery [31]. Such a high requirement for the accuracy of intraoperative localization was not necessary in UBE because the junction of the spinous process and the vertebral plate could be distinguished by the surgeon’s sense of touch, and lateral fluoroscopy was needed only once during preoperative positioning.

## 5. Limitation

This study was a single-center retrospective study with several limitations, including a small sample size and relatively short follow-up times. It also had retrospective bias during data collection. Additional randomized controlled trials with larger sample sizes are needed to confirm the effectiveness and safety of UBE and PE in the treatment of CSR.

## 6. Conclusions

In conclusion, UBE and PE were both safe and effective in the treatment of CSR and were characterized by minimal trauma, no adverse impact on cervical stability, and few complications. The fluoroscopic and operation times of UBE were shorter than those of PE.

## Figures and Tables

**Figure 1 medicina-59-00437-f001:**
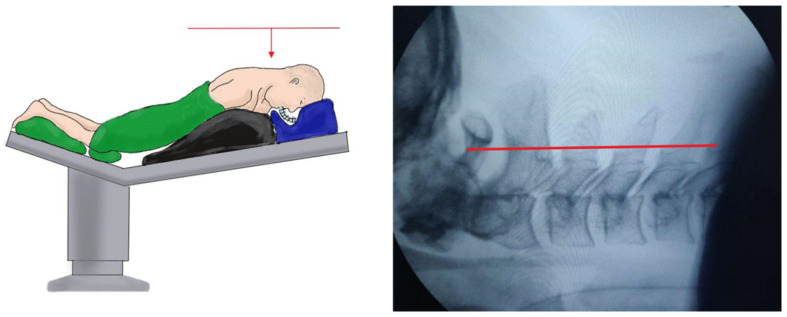
Surgical positioning and surgical posture requirements of UBE (unilateral biportal en-doscopic) and PE (percutaneous posterior endoscopic cervical discectomy) keyhole surgery.

**Figure 2 medicina-59-00437-f002:**
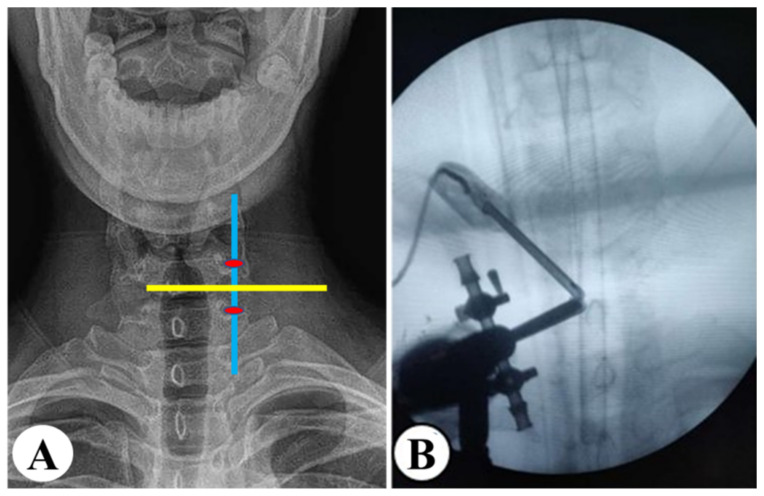
Schematic of incisions and portals. (**A**) Design of incisions. Yellow line: the horizontal line of the intervertebral space. Blue line: the midline of lateral mass. Red ovals: the viewing and working portals. (**B**) The endoscope and radiofrequency probe.

**Figure 3 medicina-59-00437-f003:**
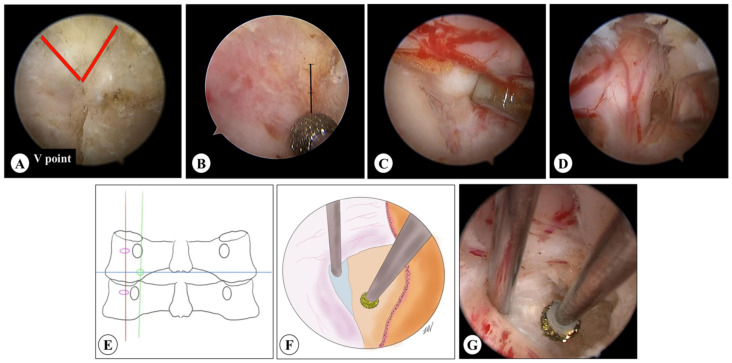
Decompression steps of UBE keyhole surgery. (**A**) The V point. (**B**) The decompression range was estimated based on the diameter of the burr. (**C**) The herniated disc. (**D**) The nerve root after decompression. (**E**) The quarterback K portal (green circle) was located at the intersection of the medial edge of the pedicle (green line) and the horizontal line of the intervertebral disc (blue line). (**F**,**G**) The nerve root was protected by a Kirschner wire.

**Figure 4 medicina-59-00437-f004:**
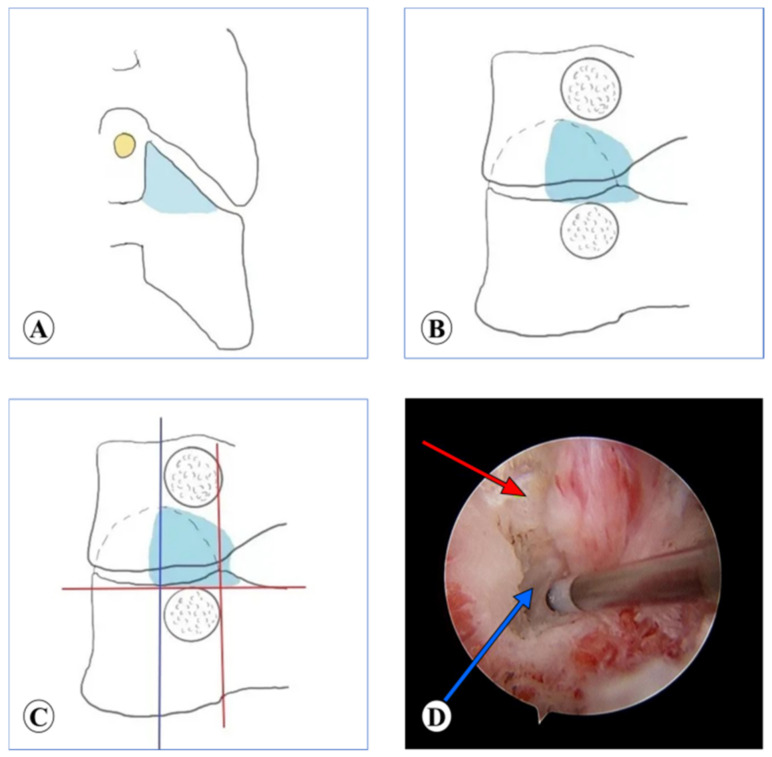
The decompression range of UBE keyhole surgery. (**A**,**B**) The decompression range in UBE keyhole surgery. (**C**,**D**) The lateral (blue arrow) and medial edge (red arrow) of the pedicle.

**Figure 5 medicina-59-00437-f005:**
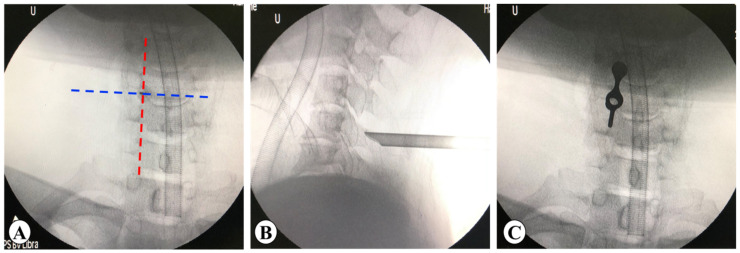
Location of target point in PE keyhole surgery. (**A**) Blue dotted line: the horizontal line of the intervertebral space. Red dotted line: the medial line of the lateral mass. (**B**,**C**) Imaging position of PE working cannula.

**Figure 6 medicina-59-00437-f006:**
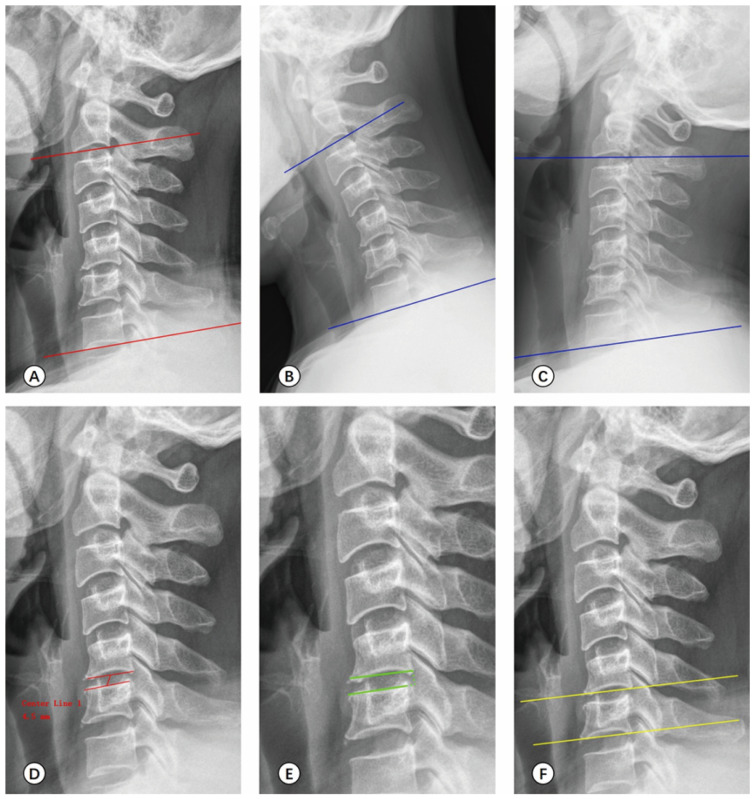
Schematic of imaging measurement. (**A**) C2-7 Cobb’s angle. (**B**,**C**) C2-7 ROM. (**D**) Intervertebral height. (**E**) Vertebral horizontal displacement. (**F**) Vertebral angular displacement.

**Figure 7 medicina-59-00437-f007:**
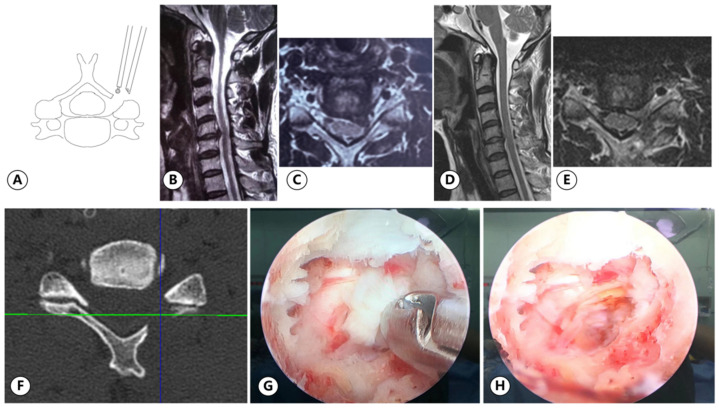
(**A**) Typical case of UBE surgery: a 51-year-old female with neck pain accompanied by radiating pain of left upper limb for five years and aggravated pain for six months. UBE operation diagram. (**B**,**C**) The preoperative MRI images. (**D**,**E**) The postoperative MRI images. (**F**) The postoperative CT image revealed the range of bony decompression. (**G**) The nucleus pulposus was removed during surgery. (**H**) The nerve root decompression was confirmed.

**Figure 8 medicina-59-00437-f008:**
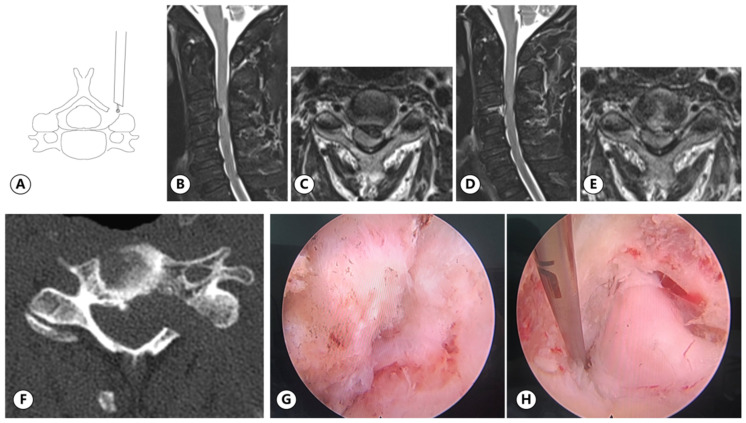
(**A**) Typical case of PE surgery: a 60-year-old male with right limb numbness and unstable walking for 2 weeks. PE operation diagram. (**B**,**C**) The preoperative MRI images. (**D**,**E**) The postoperative MRI images. (**F**) The postoperative CT image revealed the range of bony decompression. (**G**) The V point. (**H**) The nerve root decompression was confirmed.

**Figure 9 medicina-59-00437-f009:**
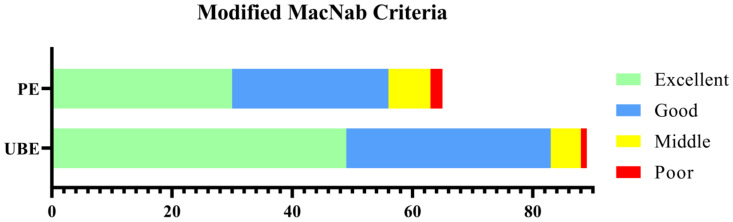
Evaluation of the outcome using modified MacNab criteria.

**Table 1 medicina-59-00437-t001:** Patient Characteristics.

	UBE	PE	*p*	*χ* ^2^	*t*
Age (years)	58.28 ± 11.94	60.10 ± 9.69	0.301		−1.038
Gender (n)					
Male	42	28	0.613	0.256	
Female	47	37			
Surgical segment					
C4/5	15	14	0.154	3.735	
C5/7	30	29			
C6/7	44	22			
Follow-up duration (months)	26.48 ± 2.22	26.58 ± 1.72	0.750		−0.320
Postoperative hospital stay (days)	6.88 ± 1.92	6.38 ± 2.24	0.156		1.428
Total hospitalization cost (thousand RMB)	24.09 ± 2.44	18.62 ± 2.87	<0.001		12.725

Unilateral biportal endoscopic (UBE), percutaneous posterior endoscopic cervical discectomy (PE).

**Table 2 medicina-59-00437-t002:** Surgical Data.

	UBE	PE	*p*	*χ* ^2^	*t*
Operation time (min)	77.48 ± 17.37	84.92 ± 21.97	0.020		−2.346
Fluoroscopic time (s)	6.76 ± 1.09	8.31 ± 1.10	<0.001		−8.649
Length of incisions (mm)	24.52 ± 2.06	11.68 ± 1.88	<0.001		39.595
Bleeding volume (mL)	55.39 ± 15.59	53.46 ± 12.50	0.198		1.293
Complications (n, %)	3 (3.37%)	4 (6.15%)	0.456 *		
Dural tear	2	1			
Nerve root injury	0	1			
Nucleus pulposus residue	1	2			
Effective rate (%)	93.26%	86.15%	0.142	2.157	

* Fisher’s precision probability test.

**Table 3 medicina-59-00437-t003:** VAS and NDI scores before and after surgery.

	Before Surgery	After Surgery				*F*	*p*
		1 Month	3 Months	6 Months	12 Months		
VAS Neck							
UBE	7.93 ± 0.69	3.28 ± 0.58	2.53 ± 0.68	2.12 ± 0.53	1.72 ± 0.45	1663.172	<0.001
PE	8.06 ± 0.83	3.14 ± 0.98	2.69 ± 0.58	2.23 ± 0.79	1.89 ± 0.69	685.597	<0.001
VAS Arm							
UBE	6.28 ± 0.94	3.60 ± 1.08	2.80 ± 0.87	2.24 ± 1.20	1.92 ± 0.80	267.264	<0.001
PE	6.35 ± 1.11	3.75 ± 0.79	2.97 ± 0.87	2.20 ± 0.73	1.86 ± 0.73	312.955	<0.001
NDI							
UBE	35.67 ± 4.24	20.69 ± 3.95	18.92 ± 4.13	17.04 ± 3.02	14.58 ± 3.09	430.917	<0.001
PE	36.25 ± 3.40	21.02 ± 4.09	19.26 ± 4.04	17.52 ± 3.02	15.18 ± 3.23	361.802	<0.001

**Table 4 medicina-59-00437-t004:** Outcomes of imaging measurements.

	Before	12 Months After Surgery	Last Follow-Up	*F*	*p*
C2-7 Cobb’s angle (°)					
UBE	16.87 ± 2.89 *^,†^	23.41 ± 4.04 *	24.46 ± 3.94 ^†^	97.527	<0.001
PE	16.66 ± 2.70 *^,†^	22.38 ± 5.73 *	23.53 ± 6.02 ^†^	39.525	<0.001
C2-7 ROM (°)					
UBE	22.60 ± 7.59 *^,†^	24.58 ± 9.58 *^,‡^	27.57 ± 9.45 ^†,‡^	6.997	0.001
PE	22.08 ± 8.23 *^,†^	25.71 ± 8.17 *	26.42 ± 9.62 ^†^	4.500	0.013
Intervertebral height of the surgical segment (mm)					
UBE	5.51 ± 0.38	5.43 ± 0.42	5.52 ± 0.40	1.202	0.303
PE	5.42 ± 0.37 *	5.61 ± 0.37 *	5.56 ± 0.40	1.865	0.159
Vertebral horizontal displacement of the surgical segment (mm)					
UBE	1.24 ± 0.28	1.25 ± 0.29	1.32 ± 0.24	1.960	0.144
PE	1.21 ± 0.29	1.26 ± 0.29	1.20 ± 0.26	0.897	0.410
Vertebral angular displacement of the surgical segment (°)					
UBE	5.39 ± 1.16	5.55 ± 1.25	5.66 ± 1.20	1.276	0.282
PE	5.27 ± 1.34	5.39 ± 1.32	5.29 ± 1.29	0.125	0.883

*,^†^,^‡^
*p* < 0.05: Pairwise comparison in different time.

## Data Availability

The original contributions presented in the study are included in the article, further inquiries can be directed to the corresponding author.

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
