# Peer review of "Comparison of Outcomes between Unilateral Biportal Endoscopic and Percutaneous Posterior Endoscopic Cervical Keyhole Surgeries"

_medicina, 2023, doi:10.3390/medicina59030437_

Round 1

Reviewer 1 Report

The authors investigated differences in clinical data between patients underwent unilateral biportal endoscopic (UBE) and percutaneous posterior endoscopic cervical disectomy (PE) for cervical spondylotic radiculopathy. They found that the fluoroscopy and surgical time of UBE were shorter than those of PE. Both UBE and PE were effective and no significant difference was observed in the postoperative hospital stay, bleeding volume, visual analogue scale, neck disability index, the modified MacNab criteria, complication rate and radiographic parameters between the two groups.

The data is interested to spinal surgeons and neurosurgeons, as well as researchers in the field. There are several concerns. I would like to advise the authors to include these points as new paragraphs in the manuscript.

Materials and methods

1.     Could you how you chose the surgical procedure?

2.     Could C7 vertebral body be seen clearly in all cases when C2-7 ROM was measured? If the shoulder obscured C7 vertebra, how did you measure?

MM and results

3.     Could you describe postoperative treatments included use of painkillers? The postoperative therapy might effect on clinical outcomes.

Author Response

Reviewer 1

Dear professor

Thank you very much for your professional review and help.

Response 1:Could you how you chose the surgical procedure?

Thanks to the professor for your question, which is a very good question.For a long time, both UBE and PE have been used to deal with cervical spondylosis caused by para central herniation of cervical vertebrae, but also used for cervical spondylosis with foramina or lateral recess stenosis, and also for single-segment cervical stenosis caused by hypertrophy of ligamentum flavum. However, with the increase of our UBE technology proficiency, more and more UBE technology with better operational flexibility should shorten the operation time and reduce the number of patient transmission times.

Response 2: Could C7 vertebral body be seen clearly in all cases when C2-7 ROM was measured? If the shoulder obscured C7 vertebra, how did you measure?

This is a very professional question, thank you for your question. In the measurement of the C7-2 ROM, we did encounter a shoulder occlusion of the C7 vertebrae, but the number was minimal.For this particular case, we will use CT scan reconstruction before the measurement.

Response 3:Could you describe postoperative treatments included use of painkillers? The postoperative therapy might effect on clinical outcomes.

Thank you very much for your question. For the method of postoperative analgesia, the two groups, local infiltration of lidocaine incision, plus three days of oral non-steroidal analgesia. All postoperative patients can achieve good analgesic effect.

Reviewer 2 Report

The authors tried to describe superiority of the  unilateral biportal endoscopic (UBE) in comparison between UBE and percutaneous posterior endoscopic cervical discectomy (PE) key-hole surgery. The article documented superiority of UBE very well based on the scientific evidences.

However I want to suggest two points.

1: In the legend of Figure 6 and Figure 8,  the authors should explain  the differences between UBE and PE, using images and photos to understand easily. It is difficult to understand without explanation between the approach of UBE and PE.

2: I think that as the trouble in operative, dural tear 2 cases in UBE and nerve roots injury one case both  UBE and PE is big problems. These trouble and complication by operative technique must not be  occur in functional surgery. So the authors described the situation of these troubles have occurred, prevent, and the methods of rescue.

After confirmation of revision according the suggestion of reviewers, this article should be consider to accept.

Author Response

Reviewer 2

Dear professor

Thank you very much for your professional review and valuable suggestions.

Response 1:In the legend of Figure 7 and Figure 8,  the authors should explain  the differences between UBE and PE, using images and photos to understand easily. It is difficult to understand without explanation between the approach of UBE and PE.

Thank you very much for your suggestion. We are sorry for failing to show it completely in Figure 7 and 8. We have added the distinguishing pictures between UBE and PE operations in the original text.

Response 2:I think that as the trouble in operative, dural tear 2 cases in UBE and nerve roots injury one case both  UBE and PE is big problems. These trouble and complication by operative technique must not be  occur in functional surgery. So the authors described the situation of these troubles have occurred, prevent, and the methods of rescue.

Thank you very much for your professional evaluation, and we also regret the results of the operation.In all the procedures we performed, two dural sac tears occurred, due to the close adhesion of the ligamentum flavum and the dural sac, and the inevitable small tear of the dural sac during the intraoperative decompression.After surgery, there was no negative pressure drainage device, and cervical braking was observed in bed. Fortunately, both patients did not have any adverse symptoms after surgery. For this "close adhesion", we believe that the exposure can be done by layer by layer stripping of the ligament, rather than the attempted overall removal.

Unfortunately, during our early PE surgery, there was one case of nerve root injury symptoms.Due to the early single-channel endoscopic technique, the field of vision and operating space during the operation are limited, and the nerve root is pulled for a long time to decompress the ventral side of the nerve root during the decompression process, resulting in a decline in the strength of the extensor dorsi of the thumb after the operation.Through nutritional nerve, hyperbaric oxygen treatment and rehabilitation exercise, the patient's thumb back extensor muscle strength was finally recovered.

Round 2

Reviewer 2 Report

The authors properly revised the manuscript according to suggestions of reviewer. Because of the explanation of UBE and PE, and the differences became to be clear, so the readers will easily understand.